# Electron Donor and Acceptor Influence on the Nonlinear Optical Response of Diacetylene-Functionalized Organic Materials (DFOMs): Density Functional Theory Calculations

**DOI:** 10.3390/molecules24112096

**Published:** 2019-06-02

**Authors:** Muhammad Khalid, Riaz Hussain, Ajaz Hussain, Bakhat Ali, Farrukh Jaleel, Muhammad Imran, Mohammed Ali Assiri, Muhammad Usman Khan, Saeed Ahmed, Saba Abid, Sadia Haq, Kaynat Saleem, Shumaila Majeed, Chaudhary Jahrukh Tariq

**Affiliations:** 1Department of Chemistry, Khwaja Fareed University of Engineering & Information Technology, Rahim Yar Khan 64200, Pakistan; bakhatali@gmail.com (B.A.); Farrukh1002@hotmail.com (F.J.); Jamsaeed41@gmail.com (S.A.); sabaabid206@gmail.com (S.A.); muskanhaq557@gmail.com (S.H.); kainat.saleem221@yahoo.com (K.S.); shumaila.ryk.2016@gmail.com (S.M.); jahrukh@gmail.com (C.J.T.); 2Department of Chemistry, University of Education Lahore, D.G. Khan Campus, Dera Ghazi Khan 32200, Pakistan; riaz.hussain@ue.edu.pk; 3Institute of Chemical Sciences, Bahauddin Zakariya University, Multan 60800, Pakistan; 4Department of Chemistry, Faculty of Science, King Khalid University, Abha 61413, P.O. Box 9004, Saudi Arabia; imranchemist@gmail.com (M.I.); aliabdullahasirri@gmail.com (M.A.A.); 5Department of Applied Chemistry, Government College University, Faisalabad 38000, Pakistan; usman.chemistry@gmail.com

**Keywords:** diacetylene-based compounds, NLO properties, quantum chemical study, density functional theory (DFT)

## Abstract

Herein, we report the quantum chemical results based on density functional theory for the polarizability (*α*) and first hyperpolarizability (*β*) values of diacetylene-functionalized organic molecules (DFOM) containing an electron acceptor (A) unit in the form of nitro group and electron donor (D) unit in the form of amino group. Six DFOM **1**–**6** have been designed by structural tailoring of the synthesized chromophore 4,4′-(buta-1,3-diyne-1,4-diyl) dianiline (**R**) and the influence of the D and A moieties on *α* and *β* was explored. Ground state geometries, HOMO-LUMO energies, and natural bond orbital (NBO) analysis of all DFOM (**R** and **1**–**6**) were explored through B3LYP level of DFT and 6-31G(d,p) basis set. The polarizability (*α*), first hyperpolarizability (*β*) values were computed using B3LYP (gas phase), CAM-B3LYP (gas phase), CAM-B3LYP (solvent DMSO) methods and 6-31G(d,p) basis set combination. UV-Visible analysis was performed at CAM-B3LYP/6-31G(d,p) level of theory. Results illustrated that much reduced energy gap in the range of 2.212–2.809 eV was observed in designed DFOM **1**–**6** as compared to parent molecule **R** (4.405 eV). Designed DFOM (except for **2** and **4**) were found red shifted compared to parent molecule **R**. An absorption at longer wavelength was observed for **6** with 371.46 nm. NBO analysis confirmed the involvement of extended conjugation and as well as charge transfer character towards the promising NLO response and red shift of molecules under study. Overall, compound **6** displayed large <*α*> and *β_tot_*, computed to be 333.40 (a.u.) (B3LYP gas), 302.38 (a.u.) (CAM-B3LYP gas), 380.46 (a.u.) (CAM-B3LYP solvent) and 24708.79 (a.u.), 11841.93 (a.u.), 25053.32 (a.u.) measured from B3LYP (gas), CAM-B3LYP (gas) and CAM-B3LYP (DMSO) methods respectively. This investigation provides a theoretical framework for conversion of centrosymmetric molecules into non-centrosymmetric architectures to discover NLO candidates for modern hi-tech applications.

## 1. Introduction

In recent years, organic nonlinear optical (NLO) materials have been the focus of intense research because of their promising functions in the field of optoelectronic technologies for signal processing, optical switching, telecommunication and information storage [1,2,3]. The reasons for the focus of organic compounds involves their facile synthesis, low cost, and structural tailoring which enables the chemical tuning of their structures for desired NLO properties. NLO properties of materials are governed by intramolecular charge transfer (ICT) which mainly originates from donor (D) to acceptor (A) moieties via π-conjugated bridges [4,5,6,7]. The design of high performance NLO materials involves appropriate donor-π-conjugated bridge-acceptor (D-π-A) systems that can be tailored through structural modification of the π-conjugated bridge, D or A substituents [8,9,10,11,12,13,14,15,16,17]. Materials having diacetylene complexes have attracted much interest in photocatalysis, optics, electronics, photo and thermochromism [18,19,20]. Therefore polymers and small molecules having diacetylene moieties in their backbone have been widely explored [21]. For these reasons, huge efforts have been made to suggest extremely valuable diacetylene functionalized organic materials (DFOMs) for hi-tech applications. Creation of these compounds involves the addition of appropriate substituents at suitable active positions to improve the NLO response [22]. In 2017 Pachfule et al. [23] utilized the diacetylene-based compound 4,4′-(buta-1,3-diyne-1,4-diyl)dianiline (BDDA) for the synthesis of the diacetylene-bridged covalent organic framework (COF) TP-BDDA via an acid-catalyzed solvothermal reaction. In the BDDA architecture, two phenyl rings are connected with each other through a diacetylene bridge and two amino groups are present at both ends of phenyl rings leading to a centrosymmetric configuration. To the best of our knowledge and literature survey, efficient theoretical investigations of BDDA for their potential NLO properties have not been reported so far. For that reason, a computational study has been planned to estimate NLO properties of such kinds of materials. In the present computational investigation, the original compound BDDA is named as **R** and a series of compounds **1**–**6** are designed by substituting the amino group of BDDA with nitro groups and changing the numbers of amino and nitro groups at both ends of the phenyl rings in the BDDA structure. 

The results related to density functional theory (DFT) and time dependent density functional theory (TDDFT) have been calculated for the estimation of absorption spectra, electronic properties and first hyperpolarizability values of **R** and the recently designed compounds **1**–**6** with the assurance that this research can provide new approach for the design of novel diacetylene-based compounds. This theoretical investigation is not only significant for representing the first estimation of the NLO properties of diacetylene-functionalized organic NLO materials but also to investigate the influence of various substituents on NLO response properties. It is expected that this effort will offer a medium for experimental researchers to produce diacetylene-functionalized organic NLO compounds.

## 2. Computational Procedure

All theoretical calculations have been conducted on Gaussian 09 program package [24] Avogadro [25], Gauss View 5.0 [26] package, and Chemcraft [27] which were very helpful for obtaining different graphic views of **R** and **1**–**6**. DFT and TDDFT calculations were performed for the estimation of electronic structures, absorption spectra and NLO properties of the diacetylene-based compounds. Ground state geometry optimization of **R** and **1**–**6** has been carried in gas phase using B3LYP level of theory and basis set like 6-31G(d,p) have been employed for the screening of organic compounds [28,29]. By using these same functional and basis set, frequency analysis was carried out to confirm the nature of optimized molecules [30,31]. During the frequency calculations, no imaginary frequency was observed which indicated the success of the true geometry optimization. Absorption spectral analysis is studied by TDDFT using 6-31G(d,p) basis set and CAM-B3LYP [32] functional which has been proved to be appropriate for charge transfer type excitations, was chosen to stimulate the absorption spectra [33,34,35,36].The conventional global hybrid DFT functionals, like B3LYP, tend to overestimate first and second hyperpolarizabilities, and in some cases can provide incorrect trends even within structurally related series [16,17]. Therefore, to get an idea about the correct trends that are recovered using long-range corrected functionals, calculations of polarizabilities and first hyperpolarizabilities have been performed at the CAM-B3LYP level of theory and results are compared with those obtained using the B3LYP method. From the Gaussian output file, six *α_xx_*, *α_yy_*, *α_zz_*, *α_xy_*, *α_xz_*, *α_yz_* (linear polarizability tensors) and ten *β_xxx_*, *β_xyy_*, *β_xzz_*, *β_yyy_*, *β_xxy_*, *β_yzz_*, *β_zzz_*, *β_xxz_*, *β_yyz_*, *β_xyz_* (hyperpolarizability tensors) along *x*, *y* and *z* directions have been obtained. Since the values of the polarizabilities (*α*) and first hyperpolarizability (*β*) of Gaussian 09 program package output are reported in atomic units (a.u.), the calculated values can be transformed into electrostatic units (esu) using conversion equations (for *α*: 1 a.u. = 0.1482 × 10^−24^ esu; for *β*: 1 a.u. = 8.6393 × 10^−33^ esu) [37,38,39]. From these tensors, magnitude of average polarizability <*α*> and first hyperpolarizability (*β_tot_*) is estimated employing Equations (1) and (2), respectively [40]:
(1)$$<\alpha >\;=1/3(\alpha_{xx}+\alpha_{yy}+\alpha_{zz})$$
(2)$$\beta_{tot}={\left[{\left(\beta_{xxx}+\beta_{xyy}+\beta_{xzz}\right)}^{2}+{\left(\beta_{yyy}+\beta_{xxy}+\beta_{yzz}\right)}^{2}+{\left(\beta_{zzz}+\beta_{xxz}+\beta_{yyz}\right)}^{2}\right]}^{1/2}$$

## 3. Results and Discussion

### 3.1. Designing of Molecular Models

The present quantum chemical investigations were performed for the theoretical design and exploration of the NLO properties of diacetylene-based substituted molecules. The reference compound **R** is a centrosymmetric molecule with two phenyl rings connected with each other through a diacetylene bridge and two amino groups are present at both ends of the phenyl rings. It is well understood from the literature that NLO properties originate from non-centrosymmetric architectures. Therefore, a theoretical design of the **R** molecule has been made to obtain non-centrosymmetric DFOMs **1**–**6** by substituting one amino group of **R** with nitro groups and changing the numbers of amino and nitro groups at both ends of phenyl rings in the **R** structure. In **1**, the *para*-amino group of **R** is replaced with a nitro group and one more nitro group is substituted at the *ortho*- position with respect to the central bridge of **R**. Compound **2** is designed by replacing **the** amino group of **R** with a nitro group and the addition of an amino group at the *meta*- position adjacent to the already present amino group. By substituting the nitro group at the *ortho*- position along the side of the already present nitro group in **2** compound **3** was designed. Compound **4** contains three amino groups on the same side benzene ring and one nitro group on benzene ring on the opposite side of the central bridge. The substitution of two nitro groups on the benzene ring on one side and three amino rings on the oppositely placed benzene ring results in compound **5**. In the case of **6**, three amino groups are present on one side and three nitro groups are substituted on the oppositely placed benzene ring. In this way, six new DFOMs **1**–**6** have been designed using the best possible combinations and their structures are given in Figure 1. DFT and TDDFT computations are performed to decipher how electron D and A moieties manipulate the spectral and NLO response properties of DFOM. In this standpoint, frontier molecular orbital (FMO) insight, polarizability (*α*), hyperpolarizability (*β*), absorption wave length, light harvesting efficiency (LHE) and molecular electrostatic potential (MEP) surfaces were calculated.

### 3.2. Electronic Structure

The results of frontier molecular orbital (FMO) calculations are significant parameters for judging the chemical stability, optical and electronic properties of molecules [41]. Physicists and chemists prefer FMOs (HOMO and LUMO) for exploring various perspectives of molecules including electronic features, optical properties, charge transfer, molecular interactions and reactivity. In general, HOMO has the capability to donate electrons and LUMO has the capability of accepting electrons [42]. The evaluation of global reactivity parameters like softness, hardness, chemical reactivity and dynamic stability of the molecules under study can be worked out through band gap (E_gap_ = E_LUMO_ − E_HOMO_) value which is a decisive factor for such studies [43]. The larger values of energy gaps in molecules make them less reactive, kinetically more stable and they resist electronic configuration changes, thus becoming hard molecules. In contrast, less stable, soft and more reactive compounds possess relatively smaller energy gaps. Less band gap molecules are easily polarized and assumed to be efficiently participating in qualitative calculation of NLO response. Assuming all these considerations, E_LUMO_, E_HOMO_ and band gap values of **R** and **1**–**6** are calculated and results are presented in Table 1.

It can be observed from Table 1 that the calculated values for E_HOMO_ and E_LUMO_ energy levels of **R** are computed to be −5.097 and −0.692 eV, respectively, with a band gap value of 4.405 eV. This showed that the highest energy gap value among all investigated compounds was observed for **R**. Due to the substitution of electron D and A moieties in **1**−**6**, the energy gap starts diminishing. Energy gap values become narrow in **1**–**6** and reduced to 2.212 eV in **6**. It represents the minimum band gap value among all the compounds under investigation. The energy gap of the designed compounds **1**–**6** is much less as compared to parent compound **R**. Due to the smaller energy gap, **1**–**6** are less stable, more reactive, can have larger wavelengths and are more easily polarizable than **R**. This implies that **1**–**6** would show excellent NLO responses as compared to **R**. The calculated band gaps of **R** and **1**–**6** increase in the following order: **6** < **3** < **5** < **1** < **4** < **2** < **R**. A pictorial demonstration of the distribution model of the respective HOMO and LUMO is shown in Figure 2. 

### 3.3. Natural Bond Orbitals (NBO) Analysis

The intra- and intermolecular interactions between the electron deficient and electron rich counterparts, conjugative interactions and charge transfer associated with molecules can be best understood through NBO analysis [44]. The second order perturbation theory can be used to determine the stabilization energy *E*^(2)^ associated to donor-acceptor interactions. The donor is represented by NBO (*i*) and acceptor is represented by NBO (*j*) during NBO analysis and the stabilization energy associated with electronic movements between acceptor and donor is represented by *E*^(2)^ as:(3)$$E^{\left(2\right)}=q_{i}\frac{{\left(F_{i,j}\right)}^{2}}{\varepsilon_{j}-\varepsilon_{i}}$$

The term *q_i_* in the above equation is used for orbital occupancy, *ε_i_*, *ε_j_* are associated to diagonal NBO Fock matrix elements, whereas *F_i,j_* is used to represent off-diagonal NBO Fock matrix elements respectively. The optimized Cartesian coordinates of compounds are presented in Appendix A (see the Appendix A) whereas NBO results of investigated compounds are presented in Appendix A, respectively.

The most credible transitions observed such as: π(C_5_–C_6_)→π*(C_3_–C_4_), π(C_11_–C_12_)→π*(C_3_–C_4_), π(C_5_–C_6_)→π*(C_1_–C_2_), π(C_5_-C_6_)→π*(C_1_–C_2_), π (C_5_–C_6_)→π*(N_21_–O_23_), π(C_1_–C_2_)→π*(N_21_–O_23_) and π(C_26_–C_34_)→π*(C_31_–C_32_) are 25.41, 219.88, 224.30, 224.40, 28.49, 25.93 and 22.40 kcal/mol for **R**, **1**, **2**, **3**, **4**, **5** and **6**, respectively. Some other transitions in **R** representing conjugation are π(C_1_–C_2_)→π*(C_3_–C_4_), π(C_1_–C_2_)→π*(C_5_–C_6_), π(C_3_–C_4_)→π*(C_1_–C_2_), π(C_3_–C_4_)→π*(C_5_–C_6_), π(C_5_–C_6_)→π*(C_1_–C_2_), π(C_15_–C_17_)→π*(C_16_–C_18_) and π(C_15_–C_17_)→π*(C_20_–O_22_) with stabilization energy values found to be 14.95, 21.98, 22.66, 17.68, 15.53, 22.66 and 17.68 kcal/mol, respectively. Similarly, transitions such as π(C_1_–C_2_)→π*(C_3_–C_4_), π(C_1_–C_2_)→π*(C_5_–C_5_), π(C_3_–C_4_)→π*(C_1_–C_2_), π(C_3_–C_4_)→π*(N_30_–O_32_), π(C_5_–C_6_)→π*(C_1_–C_2_), π(C_5_–C_6_)→π*(C_2_–C_3_) and π(C_5_–C_6_)→π*(N_27_–O_28_), also show high stabilization energy values computed to be 22.82, 17.42, 13.96, 23.81, 21.17, 15.47 and 23.79 kcal/mol, π(C11–C12)→π*( C3–C4), π(C_3_–C_4_)→π*(C_9_–C_10_), π(C_5_–C_6_)→π*(C_1_–C_2_), π(C_10_–C_11_)→π*(C_12_–C_13_), and π(C_1_–C_2_)→π*( C_3_–C_4_) also show very high stabilization energies of 219.88, 27.82, 29.46, 24.93 and 25.74 kcal/mol, respectively, in compounds **1**, **2**, **3**, **4**, **5** and **6**, respectively (see Appendix A). These are the largest values among all the stabilization energies. The study of π→π* interactions is an important tool to establish the existence of conjugation and as well as the phenomenon of charge transfer in compounds under investigation. Transitions such as: π(C_15_–C_17_)→π*(C_15_–C_17_), π(C_11_–C_12_)→π*(C_13_–C_14_), π(C_9_–C_10_)→π*(C_11_–C_12_), π(C_12_–C_13_)→π*(C_10_–C_11_), π(N_30_–O_31_)→π*(N_30_–O_31_) and π(N_33_–O_34_)→π*(N_33_–O_34_) consisting of 0.94,5.72, 7.62,12.77, 7.45 and 7.42 kcal/mol stabilization energies are the smallest ones found in **R**, **1**, **2**, **3**, **4**, **5** and **6**, respectively. The weak interactions between the electron acceptor and donor moieties lead to the lowest energy. Contrary to π→π* transitions, the interactions due to σ→σ* transitions are based on very weak donor (σ)-acceptor (σ*) interactions as compared to π→π* transitions and such interactions are very significant for the investigated compounds, leading to smaller stabilization energy values. Transitions such as σ(C_3_–C_11_)→σ*(C_11_–C_12_), σ(C_12_–H_13_)→σ*(C_13_–C_14_), σ(C_12_–C_13_)→σ*(C_13_–C_14_), σ(C_11_–C_12_)→σ*(C_12_–C_13_), σ(C_10_–C_11_)→σ*(C_11_–C_12_), σ(C_10_–C_11_)→σ*(C_11_–C_12_) and σ(C_10_–C_11_)→σ*(C_11_–C_12_) contains 11.01, 16.82,16.82, 16.12, 16.01, 15.25 and 15.22 kcal/mol energy values in compounds **R**, **1**, **2**, **3**, **4**, **5** and **6,** respectively, showing reasonable stabilization energy values among all σ→σ* interactions. Whereas, transitions like σ(C_5_–C_10_)→σ*(C_5_–C_6_), σ(C_1_–H_1_)→σ*(C_1_–C_6_), σ(C_2_–C_3_)→σ*(C_3_–C_11_), σ(C_9_–C_10_)→σ*(C_10_–C_11_), σ(C_9_–C_10_)→σ*(C_10_–C_11_) and σ(C_9_–C_10_)→σ*(C_10_–C_11_) are having least stabilization energy values 0.70, 0.84, 5.32, 8.42, 8.74 and 8.67 kcal/mol in **R**, **1**, **2**, **3**, **4**, **5** and **6**, respectively.

The similar type of interactions is observed in accordance to resonance. For example, LP(O_29_)→σ*(N_27_–N_28_), LP(O_26_)→σ*(C_6_–N_24_), LP(O_26_)→π*(N_6_–O_25_), LP(O_33_)→π*(N_32_–O_34_), LP(O_22_)→π*(N_6_–O_23_), LP(O_32_)→π*(N_30_–O_31_), LP(O_31_)→π*(N_30_–O_32_) and LP(O_52_)→σ*(N_50_–S_53_) produces 28.55, 113.05, 160.05, 181.53, 160.28, 164.32, 166.59 and 18.96 kcal/mol in **R**, **1**, **2**, **3**, **4**, **5** and **6**, respectively. These are the highest values among all. While LP(N_25_)→σ*(C_5_–C_6_), LP(N_27_)→π*(C_20_–C_22_), LP(N_27_)→σ*(C_13_–C_14_), LP(O_33_)→π*(C_4_–N_32_), LP(N_24_)→π*(C_16_–C_20_), LP(N_27_)→π*(C_16_–C_20_) and LP(O_52_)→σ*(N_50_–S_53_) produces 28.53, 5.08, 0.54, 5.15, 15.97, 5.17 and 5.75 kcal/mol which exhibits a very low electron donating interactions energies in **R**, **1**, **2**, **3**, **4**, **5** and **6**, respectively. The intermolecular charge transfer, presence of extended conjugation and hyperconjugative interactions of the molecule under study can be easily evaluated on the basis of above discussion. Therefore, the preceding discussion validates that the designed compounds would be excellent NLO materials due to successful migration of electrons from donor part to acceptor crossing π-bridge.

### 3.4. Nonlinear Optical (NLO) Properties

The versatile utility of NLO materials in telecommunication sector and optoelectronic technologies opens up the area of immense research towards probing excellent NLO materials. The scientists working in the experimental and computational fields of material sciences, chemistry and physics are playing their part towards the development of NLO materials through collaborating endeavors on account of their utilization in higher data rates, electro-optic modulation for data storage, better optical signal processing, harmonic generation and frequency mixing in valuable technologies of optical communication [45,46,47]. The quantum chemical calculations involving the concepts related to polarizability (*α*) and hyperpolarizability (*β*) are very helpful in understanding the structure–property relations of a molecule related to its NLO character. The electronic properties of a molecule are directly related to the strength of the optical response and such types of properties are usually in agreement with *α* and *β.* Theoretically, NLO properties have been calculated mostly in gas phase [48], so the NLO properties of **R** and **1**–**6** can be easily evaluated by assessing the NLO responses. In order to observe the manipulation in NLO properties of **R** and **1**–**6**, on account of D and A units, *α* and *β* values and their major contributing tensors have been measured in the gas phase at B3LYP/6-31G(d,p) level of theory and results are displayed in Table 2 and Table 5. The results of *α* and *β* calculated in gas phase are mentioned Table 3 and Table 6 and compared with those obtained from B3LYP method results mentioned in Table 2 and Table 5. Furthermore, the solvent also has a great influence over NLO properties [47,49]. The influence of the solvent has been proved practically and determined by measuring the solute–solvent interaction. To check the effect of medium polarity on *α* and *β* values, DFT computations have been performed at CAM-B3LYP level of theory in dimethylsulfoxide (DMSO; ε = 46.826) and results are collected in Table 4 and Table 7 which are compared with gas phase results obtained from B3LYP (Table 2 and Table 5) and CAM-B3LYP methods (Table 3 and Table 6).

The *α* value of **R** was found to be 248.39 (a.u.). This was the minimum measured value of average polarizability in comparison to all investigated compounds. Substitution of D and A moieties in **R** gradually increases the *α* value in **1**–**6**. The maximum measured value in this regard was observed for **6** with average polarizability value found to be 333.40 (a.u.). This is the highest value of average polarizability among all investigated compounds. The changes observed in measured values also confirmed the effectiveness of D and A units at enhancing the *α* value. In this case, following decreasing order of average polarizability was found for the compounds under investigation: **6** > **5** > **3** > **1** > **4** > **2** > **R**.

The gas phase results of investigated molecules **R** and **1**–**6** obtained from CAM-B3LYP/6-31G(d,p) level of theory indicate that the *α* value of **R** is measured as 233.24 (a.u.). This *α* value of **R** is computed as the lowest in the gas phase at the CAM-B3LYP level among all investigated compounds **R** and **1**–**6**. The highest *α* value from Table 3 is found to be for compound **6.** The results of other studied compounds indicate that substitution of D and A moieties in **R** gradually increases the *α* value in the designed molecules **1**–**6**, which also confirmed the effectiveness of D and A units at enhancing the *α* value. Overall, the results obtained from the CAM-B3LYP/6-31G(d,p) level of theory are found to be in following decreasing order: **6** > **5** > **3** > **1** > **4** > **2** > **R**. On comparing the gas phase results of B3LYP and CAM-B3LYP functionals, it can be seen that *α* value is found to have the same decreasing order (**6** > **5** > **3** > **1** > **4** > **2** > **R**) in both cases. The comparison of lowest and highest *α* values for **R** and **6** shows that CAM-B3LYP results are noticeably lower as compared to the B3LYP results, respectively. A similar trend can be seen in the case of the remaining investigated compounds where B3LYP results are found to be higher as compared to CAM-B3LYP results. 

As far as CAM-B3LYP solvent phase results are concerned, it is well evident from Table 4 that the polarizability of the title molecule is also changed with the polarity of the solvent. The solvent notably altered and increased the polarizability of the investigated molecules as compared to the gas phase results. Overall, the results mentioned in Table 4 are found to be following the decreasing order: **6** > **5** > **3** > **1** > **4** > **2** > **R** which is similar to the decreasing order obtained from the gas phase results of the B3LYP and CAM-B3LYP methods. In solvent phase, the lowest *α* values of **R** are increased from 248.39 a.u. (B3LYP) and 233.24 a.u. (CAM-B3LYP) obtained from gas phase results to
302.80 a.u. in DMSO solvent. Similarly, the highest value in solvent is noted 380.46 a.u. for compound **6**, which is also greater than the gas phase results of B3LYP and CAM-B3LYP.

Transition along x and y directions are commonly used to calculate the polarizability (along the x direction) and is explained by the following equation [50,51,52]:
(4)$$\alpha \propto \frac{{\left(M_{X}^{\mathrm{gm}}\right)}^{2}}{E_{\mathrm{gm}}}$$

In this equation, $M_{x}^{\mathrm{gm}}$ represents the transition moment whereas the denominator (E_gm_) represents the transition energy in accordance with electronic energy states (ground state and excited state) of molecule. The estimation of phase factor related to ground state and excited state was worked out with the help of transition dipole moment ($M_{x}^{\mathrm{gm}})$, a vector quantity. The electronic transitions lead to polarization which was best described through transition dipole moment. The transition dipole moment was also used to account for the interaction of electromagnetic wave of a particular polarization with the compound under study. 

The inverse relation between *α* and transition energy is presented in Equation (4) and the transition energy is directly related to the transition dipole moment. In other words, the dipole polarizability value is amplified by the value of the transition moment. In general, large hyperpolarizability values are associated to large values of $M_{x}^{gm}$ and relatively smaller values of transition energy, so the quantitative values of the hyperpolarizability of a compound can be used to judge the NLO activity of the compound. The NLO response and dipole moment of a molecule are altered by the interface of the electronic density and the external electric field [53]. Herein, the calculations with respect to B3LYP (gas), CAM-B3LYP (gas) and CAM-B3LYP (DMSO) level hyper- polarizabilities of **R** and **1**–**6** have been presented in Table 5, Table 6 and Table 7, respectively, along with the dominant contributing tensors of *β* and *β_tot_*.

The data presented in Table 5, Table 6 and Table 7 points out that the diagonal *β_xxx_* component has a major contribution towards the value of *β_tot_* among all *β* components. The calculated *β_tot_* values of **R** were found to be 110.56 (a.u.), 95.57 a.u., and 165.55 a.u., from the B3LYP (gas phase), CAM-B3LYP (gas phase) and CAM-B3LYP (DMSO) methods, respectively. These values represent the smallest values of *β_tot_* response among all compounds under study. Substitution of electron D and A units in **R** notably improved the *β_tot_* which describes the operative influence of D and A moieties in tuning the NLO response. Among all investigated compounds, designed system **6** was found to have the highest *β_tot_* values found using the B3LYP (gas), CAM-B3LYP (gas) and CAM-B3LYP (DMSO) methods, respectively. The *β_tot_* values of **1**–**6** were observed to be 107–223 times, 80–124 times and 97–151 times greater than those of the parent **R** molecule calculated through the B3LYP (gas), CAM-B3LYP (gas) and CAM-B3LYP (DMSO) methods, respectively. The *β_tot_* value decreased gradually in the order shown for given compounds: **6** > **3** > **5** > **2** > **4** > **1** > **R** B3LYP (gas), **6** > **3** > **5** > **2** > **1** > **4** > **R** CAM-B3LYP (gas) and **6** > **3** > **5** > **2** > **1** > **4** > **R** CAM-B3LYP (DMSO). These orders are found to be in very good agreement with each other, indicating that substitution of electron D and A units in **R** remarkably enhanced the *β_tot_* values obtained from different methods.

Comparison of gas phase B3LYP results with CAM-B3LYP results indicates that the *β_tot_* values obtained from the former method are found to be greater in all investigated compounds **R** and **1**–**6** as compared to those calculated with the latter method, respectively. Furthermore, a comparison between gas phase results mentioned in Table 5 and Table 6 has been made with the solvent (DMSO) results (in Table 7) which indicates that the solvent enhanced the *β_tot_* results significantly. The lowest *β_tot_* values of **R** are increased from 110.56 a.u. (B3LYP) and 95.57 a.u. (CAM-B3LYP) gas phase results to 165.55 a.u. in DMSO solvent. A similar enhancement is noted in the case of all designed compounds. For example, the highest *β_tot_* values 24708.79 a.u. (B3LYP) and 11841.93 a.u. (CAM-B3LYP) noted for compound **6** in gas phase become further increased to 25053.32 a.u. in solvent phase. It is evident from the above results that the solvent has also a great influence over NLO response properties.

Additionally for comparative NLO analysis, we have compared our results of designed molecules **1**–**6** with 4-(dimethylamino)-4′-nitrostylbene (DANS) chromophores. The comparison of designed molecules **1**–**6** results with DANS results indicates that designed molecules **1**–**6** exhibit 1.06–2.21 times, 1.41–2.20 times, and 1.28–1.99 times greater NLO response than DANS calculated through B3LYP (gas phase), CAM-B3LYP (gas phase) and CAM-B3LYP (solvent DMSO) methods, respectively. These comparative analysis results confirmed that designed molecules possess fine NLO character and would be suitable as potential NLO candidates for NLO applications. From the preceding discussion, it is well evident that the gas phase results from the B3LYP method are better as compared to CAM-B3LYP gas phase results. Furthermore, the solvent significantly affects and enhances the NLO response properties as compared to gas phase. Moreover, substitution of electron D and A units in **R** adjusted the structure of designed molecules **1**–**6** into a configuration favorable for enhancing the NLO response of **1**–**6** as compared to the parent compound **R**.

### 3.5. UV–Vis Spectra of Compounds

TDDFT computations in the gas phase were performed at the CAM-B3LYP/6-31G(d,p) level of theory to calculate the excited state absorption spectra of the investigated compounds **R** and **1**–**6**. Thus, ten lowest energy singlet to singlet transitions were computed and results wavelength associated to maximum absorption (*λ*_max_), nature of transitions, oscillator strength (*f_os_*) and computed transition energy (E_ge_) values are tabulated in Table 8.

The absorbance exhibited by the all the compounds under study lies in the visible region as shown in Table 8. The calculated λ_max_ of **R** is observed to be 267.84 nm. The lowest absorption wavelength i.e., 244.88 nm was observed for compound **2** and the maximum value of absorption wavelength i.e., 371.46 nm was observed for compound **6** which implies that **6** absorbed at higher λ_max_ than the other studied compounds. The optical efficiency of a molecule can be accounted for through another important factor i.e., light harvesting efficiency (LHE). The larger photocurrent response of a compound is usually associated to larger LHE values. Equation (5) was utilized for working out LHE of compounds under study [54] and the findings are presented in Table 8.
LHE = 1 − 10^−f^(5)

The LHE of **R** was found to be highest among all studied compounds. The spectral representations of **R** and **1**–**6** are given in Figure 3.

These results indicate that, the control of acceptor and donor units is very vital for the development of promising materials with potential NLO responses. Furthermore, D and A units influence the NLO response of DFOMs. The impact of D and A units on NLO properties can be easily applied as a tool for designing photoelectric and optical materials with better performance in modulation, data processing and optical switching. The pairing of D and A units in organic frameworks is a common practice for coupling D and A units in designing NLO compounds with moderate to strong NLO activity. Anyhow, a particular extent of D-A coupling is indispensable for achieving a significant first hyperpolarizability. However, pairing of D and A units through a bridge should not be too strong to disturb the electronic asymmetry of the compound under study.

### 3.6. Molecular Electrostatic Potential (MEP)

MEP is a three dimensional plot of the total electron density which is used to examine the physical and chemical aspects of molecules. The appropriate sites for the feasible nucleophilic and electrophilic attack are easily explained with the help of MEP. Different standard colors such as red, orange, yellow, green and blue are used in MEP diagrams to clarify the degree of electrostatic potential which decreases in the order blue > green > yellow > orange > red. Red color was used to indicate the negative potential sites encouraging electrophilic attack. On the other hand, the nucleophile loving site with most positive potential is shown by blue color. MEP can be illustrated by the following Equation (6):
(6)V(r)=∑(ZARA−r)−∫p(r′)(r′−r)dr′

In this equation, ZA represents the charge of nucleus which is located at RA and ρ(r’) indicates the electron density. MEP surfaces for all compounds are developed utilizing the optimized geometry of compounds **R**, **1**, **2**, **3**, **4**, **5** and **6**, respectively, and their pictographic display is presented in Figure 4. From Figure 4, it can be seen that the pure red color zone on MEP is present on the oxygen atoms of **1**–**6** molecules. Therefore, this region is the electron-rich zone and related to the chance of electrophilic attack. The light red color appearing on the opposite ends of the **R** molecule is due to the presence of lone pair of electrons on the nitrogen atom. Green and blue zones are present on hydrogen and some carbon atoms indicating an electron-deficient zone which implies that the interactions with nucleophiles can take place on this region. 

## 4. Conclusions

The first hyperpolarizability (*β*) values of donor-acceptor (D-A) diacetylene-based compounds depends on the extent of electronic interactions between the amino and nitro groups of the molecular system. The increased availability of mobile electrons on the diacetylene molecules also plays a vital role in increasing the polarization of the investigated systems. Moreover, the electron withdrawing power of the nitro groups of the compounds will also have an influence on the polarization of the diacetylene molecules. This study indicated that all molecular systems have planar structures. The intramolecular charge transfer transitions are relatively strong in the designed derivatives due to their planarity, and the amino and nitro groups. This will have an influence on the hyperpolarizability. The findings indicated that B3LYP results are higher as compared to CAM-B3LYP results. These designed derivatives have larger <*α*> and hyperpolarizability (*β*) values than standard systems. Especially compound **6** contains the highest <*α*> and *β_tot_* values computed to be 333.40 (a.u.) [B3LYP gas phase], 302.38 a.u. [CAM-B3LYP gas phase], 380.46 a.u. [CAM-B3LYP solvent] and 24708.79 (a.u.), 11841.93 (a.u.), 25053.32 (a.u.) measured using the B3LYP (gas phase), CAM-B3LYP (gas phase) and CAM-B3LYP (solvent DMSO) methods, respectively, among all studied compounds. The 4-(dimethylamino)-4′-nitrostylbene (DANS) chromophores comparison analysis proved that these compounds may possess fine potential and would be suitable as potential NLO candidates for NLO applications. As hyperpolarizability is due to an intramolecular charge transfer process, these compounds might also be good candidates for artificial photosynthetic processes. Our research described an addition to the paradigm for converting centrosymmetric units into non-centrosymmetric configurations by certain modifications in the structures of D and A for designing new promising NLO compounds with huge NLO responses. Experimental scientists looking for better NLO candidates should synthesize and check the proposed compounds for optoelectronic applications.

## Figures and Tables

**Figure 1 molecules-24-02096-f001:**
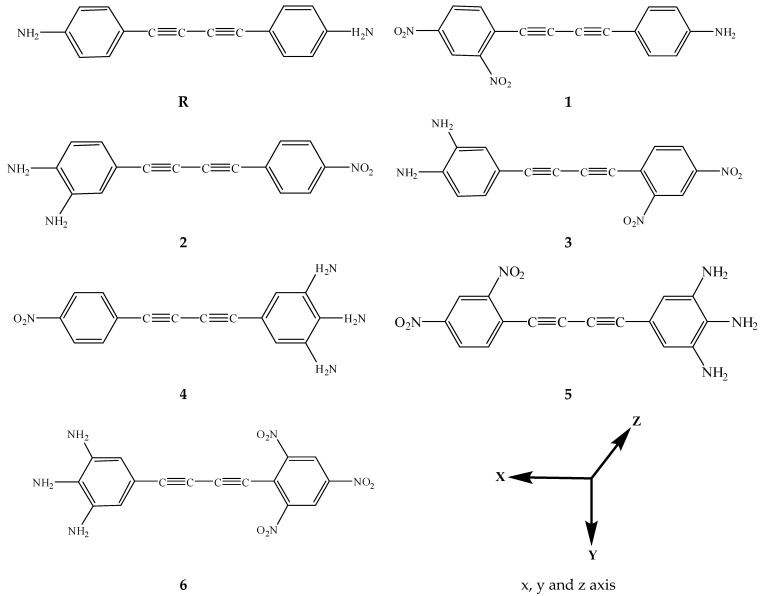
Representation of parent molecule **R** and designed compounds **1**–**6**.

**Figure 2 molecules-24-02096-f002:**
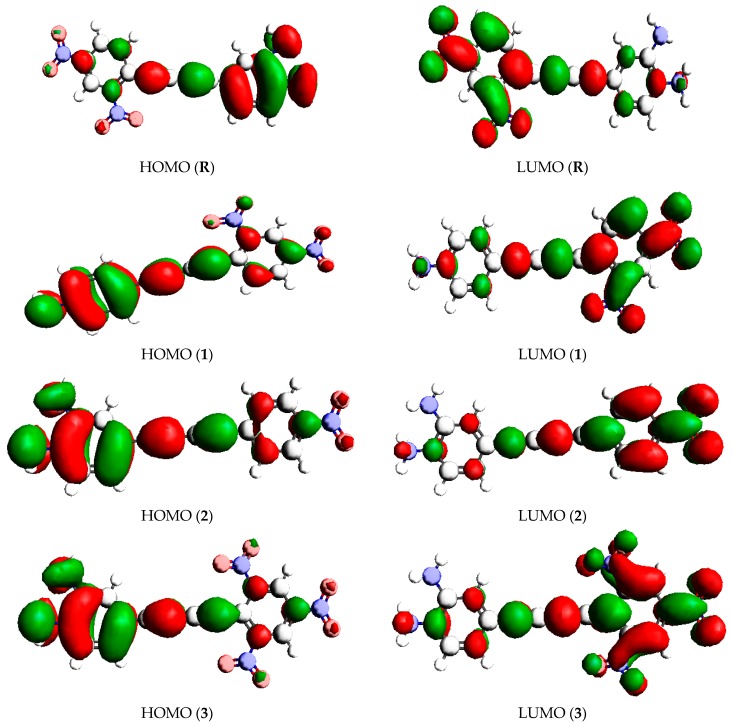
HOMOs and LUMOs of the studied compounds (**R** and **1**–**6**).

**Figure 3 molecules-24-02096-f003:**
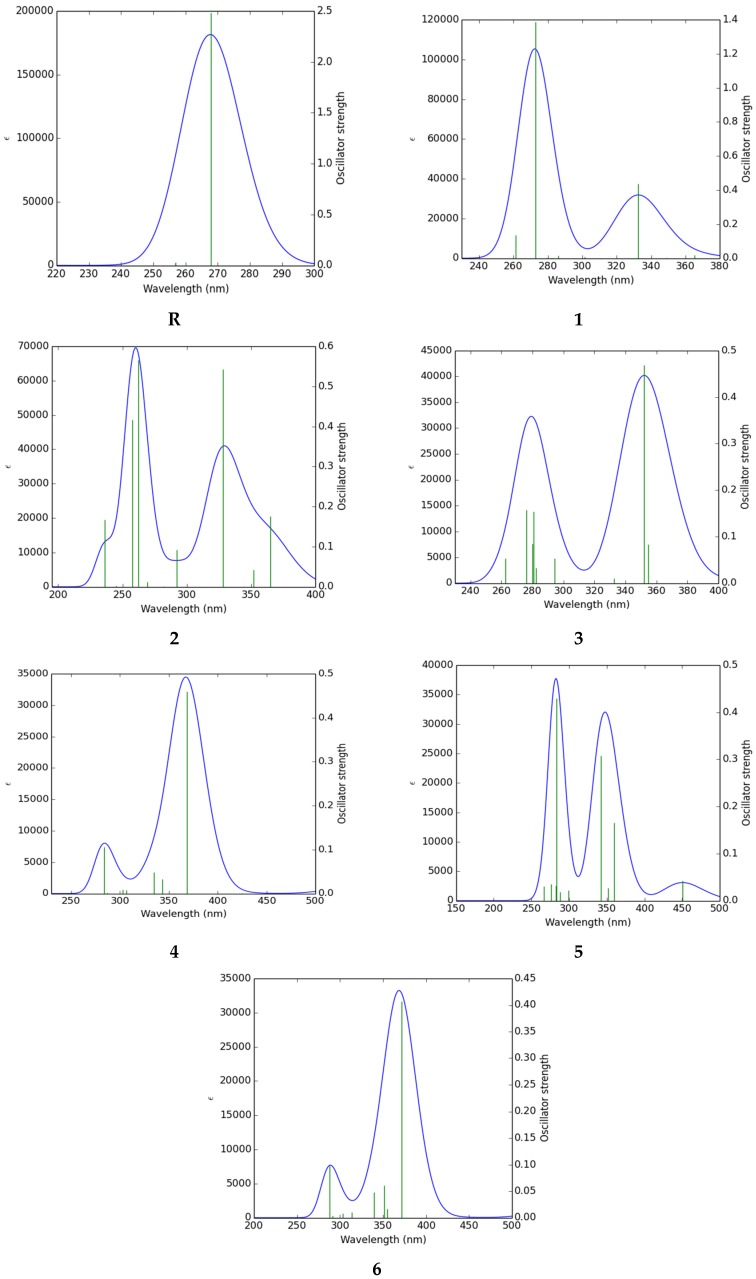
UV/Visible spectra of the studied compounds **R** and **1**–**6**.

**Figure 4 molecules-24-02096-f004:**
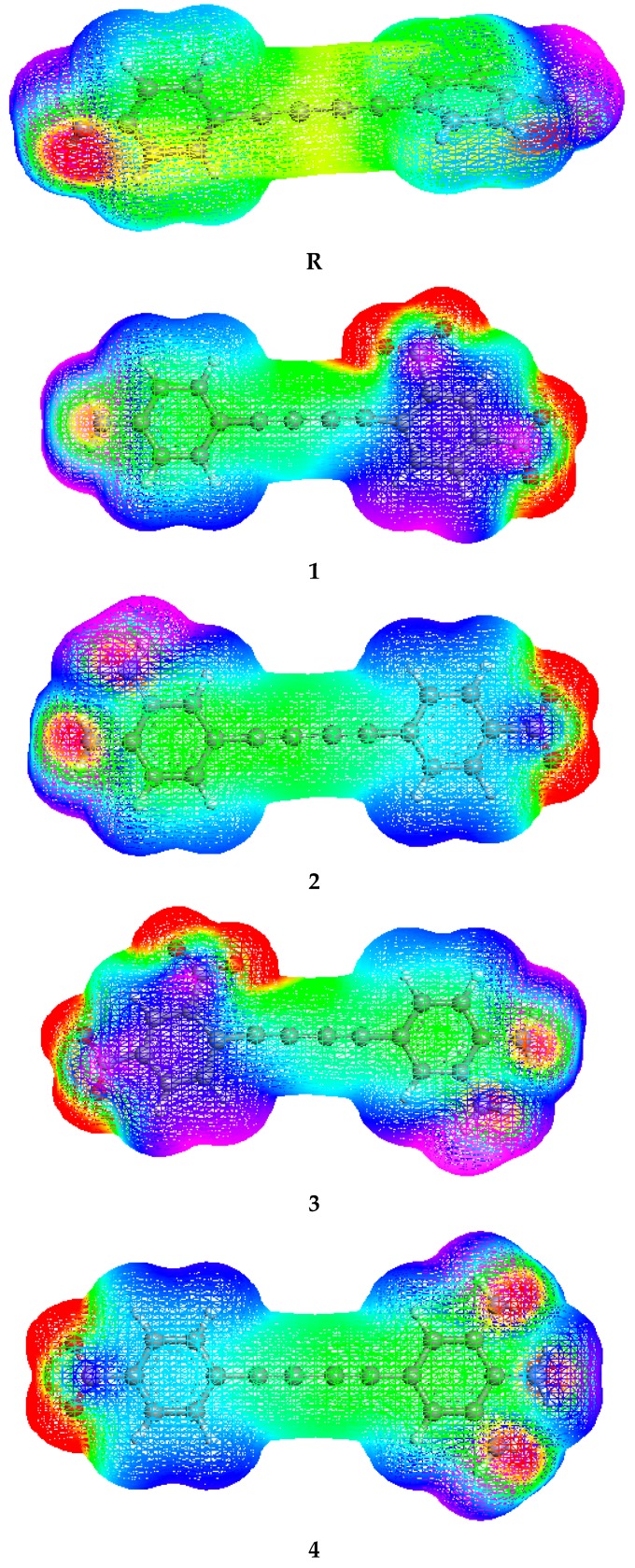
Diagrams and color scheme of the studied compounds **R** and **1**–**6**.

**Table 1 molecules-24-02096-t001:** The E_HOMO_, E_LUMO_ and energy band gap (E_LUMO_ − E_HOMO_) of the studied compounds **R** and **1**–**6**.

Systems	HOMO (E_HOMO_)	LUMO (E_LUMO_)	Band Gap ^a^
**R**	−5.097	−0.692	4.405
**1**	−5.720	−3.068	2.652
**2**	−5.364	−2.555	2.809
**3**	−5.734	−3.411	2.323
**4**	−5.226	−2.532	2.694
**5**	−5.422	−3.032	2.390
**6**	−5.590	−3.378	2.212

^a^ Band gap = E_LUMO_ − E_HOMO_.

**Table 2 molecules-24-02096-t002:** Dipole polarizabilities and major tensor (a.u.) of the studied compounds **R** and **1**–**6** in the gas phase at the B3LYP/6-31G(d,p) level of theory.

Systems	*α_xx_*	*α_yy_*	*α_zz_*	*α_(tot)_*
**R**	529.66	107.88	107.64	248.39
**1**	581.62	141.69	119.38	280.89
**2**	582.22	148.97	99.94	277.04
**3**	614.99	173.73	107.65	298.79
**4**	568.69	157.01	108.25	277.98
**5**	618.81	181.81	117.07	305.89
**6**	665.71	208.96	125.54	333.40

**Table 3 molecules-24-02096-t003:** Dipole polarizabilities and major contributing tensors (a.u.) of **R** and **1**–**6** in the gas phase at the CAM-B3LYP/6-31G(d,p) level of theory.

Systems	*α_xx_*	*α_yy_*	*α_zz_*	<*α*>
**R**	486.095	106.931	106.701	233.24
**1**	520.559	139.443	117.609	259.20
**2**	516.306	147.529	98.765	254.20
**3**	542.117	170.093	106.170	272.79
**4**	505.284	155.125	106.899	255.76
**5**	545.601	177.790	115.171	279.52
**6**	581.500	202.408	123.244	302.38

**Table 4 molecules-24-02096-t004:** Polarizabilities and major contributing tensors (a.u.) of **R** and **1**–**6** in solvent (DMSO) at the CAM-B3LYP/6-31G(d,p) level of theory.

Systems	*α_xx_*	*α_yy_*	*α_zz_*	<*α*>
**R**	631.050	138.839	138.515	302.80
**1**	663.405	185.417	153.872	334.23
**2**	661.006	196.999	124.885	327.63
**3**	697.595	227.901	134.787	353.42
**4**	634.922	205.830	135.308	325.35
**5**	695.553	237.416	146.107	359.69
**6**	747.183	270.977	123.244	380.46

**Table 5 molecules-24-02096-t005:** Computed first hyperpolarizability (*β_tot_*) and major contributing tensor (a.u.) of **R** and **1**–**6** in the gas phase at the B3LYP/6-31G(d,p) level of theory.

Systems	*β_xxx_*	*β_xxy_*	*β_xyy_*	*β_yyy_*	*β_xxz_*	*β_xzz_*	*β_(tot)_*
**R**	0.896	−97.11	−0.284	−17.333	−0.230	0.245	110.56
**1**	12064.69	−106.37	−133.46	−86.08	123.36	−86.05	11,850.78
**2**	14,073.02	249.01	−41.69	−68.84	−930.28	−0.14	14,064.43
**3**	18,549.79	12.17	23.41	−204.93	−1144.63	14.32	18,622.90
**4**	12,341.41	350.87	−10.67	−39.27	−779.17	−2.08	12,355.91
**5**	17,659.96	740.01	35.49	96.23	−1272.25	48.32	17,814.57
**6**	24,416.82	493.13	159.11	−49.25	−1570.73	78.06	24,708.79

**Table 6 molecules-24-02096-t006:** The computed second-order polarizabilities (*β_tot_*) and major contributing tensors (a.u.) of **R** and **1**–**6** in the gas phase at the CAM-B3LYP/6-31G(d,p) level of theory.

Systems	*β_xxx_*	*β_xxy_*	*β_xyy_*	*β_yyy_*	*β_xxz_*	*β_yyz_*	*β_yzz_*	*β_tot_*
**R**	0.817	−82.747	−0.263	−16.629	−0.211	0.029	3.807	95.57
**1**	7849.173	−104.490	−133.343	−67.434	112.711	55.494	−45.776	7633.57
**2**	7759.690	156.817	−43.929	−58.269	−501.671	−34.127	−52.103	7710.35
**3**	9728.750	24.417	−20.110	−157.980	−577.185	6.552	−65.935	9706.27
**4**	6711.573	204.537	−17.683	−31.298	−414.521	−0.951	−31.972	6685.71
**5**	9345.049	459.870	−9.872	63.288	−665.834	−51.395	−12.036	9376.96
**6**	11,767.968	312.903	41.306	−44.651	−731.376	−7.557	−26.778	11,841.93

**Table 7 molecules-24-02096-t007:** The computed second-order polarizabilities (*β_tot_*) and major contributing tensors (a.u.) of **R** and **1**–**6** in solvent (DMSO) at the CAM-B3LYP/6-31G(d,p) level of theory.

Systems	*β_xxx_*	*β_xxy_*	*β_xyy_*	*β_yyy_*	*β_xxz_*	*β_yyz_*	*β_yzz_*	*β_tot_*
**R**	2.435	−144.230	−0.613	−27.555	−0.652	0.054	6.248	165.55
**1**	16,631.708	−231.501	−329.823	−209.552	239.991	174.871	−149.357	16,098.68
**2**	18,099.997	497.438	−141.896	−127.900	−1132.837	−71.934	−105.848	17,967.57
**3**	21,387.589	195.527	−104.886	−447.118	−1223.590	54.592	−153.540	21,284.92
**4**	14,762.046	498.304	−70.807	−65.035	−911.474	−4.685	−66.011	14,701.46
**5**	19,985.964	1056.922	−102.896	244.535	−1418.610	−153.987	−5.156	20,001.55
**6**	25,007.981	714.79	−29.686	−109.765	−1541.897	−9.18	−60.482	25,053.32

**Table 8 molecules-24-02096-t008:** Computed transition energy (eV), wavelengths (*λ*), oscillator strengths (*f_os_*), light harvesting efficiency (LHE), transition moment ($M_{X}^{\mathrm{gm}}$ a.u.) and molecular orbital (MO) transition.

Systems	*E*(eV)	*λ*_max_ (nm)	*f_os_*	LHE	$M_{X}^{\mathrm{gm}}$ (a.u.)	MO Transition
**R**	4.6287	267.84	2.4816	0.996	−0.0371	H-1→L + 1 (47%), H→L (46%)
**1**	4.5430	272.90	1.3855	0.958	3.5581	H→L + 2 (45%), H→L + 3 (20%)
**2**	5.0628	244.88	0.0016	0.003	3.0704	H-3→L (18%), H→L (17%)
**3**	3.5237	351.83	0.4689	0.660	4.5982	H-2→L (43%), H→L (15%)
**4**	4.7820	259.26	0.8677	0.864	2.6813	H-2→L + 1 (22%), H-1→L + 1 (26%)
**5**	4.3828	282.87	0.4293	0.627	0.9673	H→L+2 (30%), H→L + 3 (17%)
**6**	3.3375	371.46	0.4071	0.608	7.4945	H-2→L (37%), H→L (17%)

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
