# Peer review of "Electron Donor and Acceptor Influence on the Nonlinear Optical Response of Diacetylene-Functionalized Organic Materials (DFOMs): Density Functional Theory Calculations"

_molecules, 2019, doi:10.3390/molecules24112096_

Round 1
Reviewer 1 Report
The manuscript by Khalid et al. presents a computational DFT study on structure and nonlinear optical (NLO) properties of a series of dipolar dyes derived from 1,4-diphenyl-buta-1,3-diyne. The manuscript is well organized (easy to follow) and the statements are supported by detailed analysis. Although the novelty of this work (including design of target structures) is rather low, it could be interesting to some experimentalists when characterizing diacetylene-derived compounds for their NLO properties. I would recommend this manuscript for publication in Molecules (or in some more specialized journal focused on computer-aided modelling), given that the following points are addressed in the revised version:
a) The cited literature is relatively poor and I miss some relevant papers on structure-property relationships for dyes displaying large NLO response (ideally combining both theory and experiment). In this regard, I suggest to add (at least) the following papers to refs. 8-13:
J. Am. Chem. Soc. 2000, 122, 1154–1160
Chem. Mater. 2006, 18, 5907–5918
J. Phys. Chem. C 2010, 114, 22289–22302
J. Org. Chem. 2010, 75, 3053–3068
b) Being aware of some drawbacks of the B3LYP method in description of the charge-transfer processes, I'm wondering why authors didn't employ the CAM-B3LYP functional also for evaluating polarizabilities and hyperpolarizabilities (this was used only for calculating excitation spectra/energies). Authors should mention that conventional "global hybrid" DFT functionals, like B3LYP, tend to overestimate first and second hyperpolarizabilities, and in some cases can provide incorrect trends even within structurally related series (see for example, J. Org. Chem. 2010, 75, 3053 and J. Phys. Chem. C 2010, 114, 22289; these papers also demonstrate how the correct trends are recovered using long-range corrected functionals). In this respect, I also suggest to perform calculations of polarizabilities and static first hyperpolarizabilities at the CAM-B3LYP level and compare the results with those obtained using the B3LYP method.
c) Interestingly, neither implicit solvent effects are discussed throughout the manuscript, even though they play a big role for structurally related CT systems (i.e. I would like to see a comparison of "gas-phase" and PCM results at least for one model compound).
d) For easier comparison with experimental data, both polarizabilities and first hyperpolarizabilites should be given in esu units (or authors should at least indicate how to convert the computed values in au units to esu units).
e) What do the authors mean by "chemical strength" ? Also some small typos should be fixed prior to resubmission, e.g.: "interactionnsare" -> "interactions are" etc.
f) If possible, please define x, y and z axis in Scheme 1 (to match the labels of tensor components given in Tables 2 and 3).
Author Response
Answer to reviewers' comments to author:
Reviewer: 1
We are deeply grateful to the reviewer for taking the extensive time to provide quite valuable comments, suggestions and encouraging evaluation. We have placed our responses (in yellow text) point-by-point for each comment. In addition, we indicated revisions in the updated manuscript by a yellow highlighter in the manuscript.
Comments to the Author:
Comments and Suggestions for Authors
The manuscript by Khalid et al. presents a computational DFT study on structure and nonlinear optical (NLO) properties of a series of dipolar dyes derived from 1,4-diphenyl-buta-1,3-diyne. The manuscript is well organized (easy to follow) and the statements are supported by detailed analysis. Although the novelty of this work (including design of target structures) is rather low, it could be interesting to some experimentalists when characterizing diacetylene-derived compounds for their NLO properties. I would recommend this manuscript for publication in Molecules (or in some more specialized journal focused on computer-aided modelling), given that the following points are addressed in the revised version:
Answer: Thanks for the positive recommendation. We are profoundly thankful to respected reviewer for considering our manuscript as acceptable after revision. We are motivated as the referee has considered this manuscript well organized (easy to follow), interesting to experimentalists and its statements are supported by detailed analysis.
a) The cited literature is relatively poor and I miss some relevant papers on structure-property relationships for dyes displaying large NLO response (ideally combining both theory and experiment). In this regard, I suggest to add (at least) the following papers to refs. 8-13:
J. Am. Chem. Soc. 2000, 122, 1154–1160
Chem. Mater. 2006, 18, 5907–5918
J. Phys. Chem. C 2010, 114, 22289–22302
J. Org. Chem. 2010, 75, 3053–3068
Answer: According to the instructions of valuable reviewer, Suggested references have been cited in the revised manuscript.
b) Being aware of some drawbacks of the B3LYP method in description of the charge-transfer processes, I'm wondering why authors didn't employ the CAM-B3LYP functional also for evaluating polarizabilities and hyperpolarizabilities (this was used only for calculating excitation spectra/energies). Authors should mention that conventional "global hybrid" DFT functionals, like B3LYP, tend to overestimate first and second hyperpolarizabilities, and in some cases can provide incorrect trends even within structurally related series (see for example, J. Org. Chem. 2010, 75, 3053 and J. Phys. Chem. C 2010, 114, 22289; these papers also demonstrate how the correct trends are recovered using long-range corrected functionals). In this respect, I also suggest to perform calculations of polarizabilities and static first hyperpolarizabilities at the CAM-B3LYP level and compare the results with those obtained using the B3LYP method.
Answer: We are profoundly thankful to respected reviewer for critical review. According to the instructions of valuable reviewer, the suggested lines along with the suggested reference have been added in the revised version. Furthermore, we have performed the calculations of polarizabilities and static first hyperpolarizabilities at the CAM-B3LYP level. The obtained results are compared with those obtained earlier using the B3LYP method. The α values are found to be with same decreasing order (6>5>3>1>4>2>Sc) in both cases. The comparison of lowest and highest α values found for Sc and 6 describe that CAM-B3LYP results (233.24 and 302.38 a.u) are noticed lower as compared to the B3LYP results of lowest and highest α values (248.39 and 333.40 a.u) respectively. The similar trend can be seen in case of remaining investigated compounds where B3LYP results are found higher as compared to CAM-B3LYP results. Furthermore, comparison of B3LYP results with CAM-B3LYP results implies that βtot values obtained from former method (B3LYP) is found to be greater in all investigated compounds (Sc and 1-6) as compared to those calculated from latter method (CAM-B3LYP) respectively. For instance, lowest and highest βtot values for Sc and 6 are found to be 110.56 and 24708.79 a.u from B3LYP results; while 95.57 and 11841.93 a.u from CAM-B3LYP method respectively.
Thus, new Tables have been added in the revised version as suggested by the respected reviewer. The discussion has been changed accordingly in the revised manuscript. The numbering of the Tables has been modified accordingly.
c) Interestingly, neither implicit solvent effects are discussed throughout the manuscript, even though they play a big role for structurally related CT systems (i.e. I would like to see a comparison of "gas-phase" and PCM results at least for one model compound).
Answer: We are deeply grateful to respected reviewer for the nice suggestion. Previously, we had performed the calculations of polarizabilities and static first hyperpolarizabilities in gas phase at the B3LYP method. Now, as per instructions of valuable reviewer, the effect of solvent on α and β values have been examined by performing DFT computations at CAM-B3LYP level of theory in gas phase and as well as in solvent dimethylsulfoxide (DMSO; ε=46.826) phase and results are compared with gas phase results obtained from B3LYP and CAM-B3LYP methods. It is well evident from results that solvent has notably altered and increased the polarizability of investigated molecules as compared to gas phase results. Overall, solvent phase results are found to be in following decreasing order: 6>5>3>1>4>2>Sc which is similar to decreasing order obtained from gas phase results of B3LYP and CAM-B3LYP methods. In solvent phase, the lowest α values of Sc are increased from 248.39 a. u (B3LYP) and 233.24 a. u (CAM-B3LYP) obtained from gas phase results to 302.80 a.u in DMSO solvent. Similarly, the highest value in solvent is noted 380.46 a.u for compound 6 which is also greater than gas phase results of B3LYP (333.40 a. u) and CAM-B3LYP (302.38 a.u).
In case of βtot values, comparison between gas phase results with solvent (DMSO) results implies that solvent significantly exaggerated the βtot results. The lowest βtot values of Sc are increased from 110.56 a. u (B3LYP) and 95.57 a. u (CAM-B3LYP) gas phase to 165.55 a. u in DMSO solvent. Similar enhancement is noted in case of all designed compounds. For example, the highest βtot values 24708.79 a. u (B3LYP) and 11841.93 a.u (CAM-B3LYP) noted for compound 6 in gas phase increased further to 25053.32 a.u in solvent phase. It is evident from above results that solvent has also great influence over NLO response property.
Thus, new Tables have been added in the revised version as suggested by the respected reviewer. The discussion has been changed accordingly in the revised manuscript. The numbering of the Tables has been modified accordingly.
d) For easier comparison with experimental data, both polarizabilities and first hyperpolarizabilites should be given in esu units (or authors should at least indicate how to convert the computed values in au units to esu units).
Answer: According to the guidelines of valuable reviewer, following lines describing the conversion of a.u into esu units has been added in revised manuscript. “Since the values of the polarizabilities (α) and first hyperpolarizability (β) of Gaussian 09 program package output are reported in atomic units (a.u.), the calculated values can be transformed into electrostatic units (esu) using conversion equations (for α: 1 a.u. = 0.1482 x10-24 esu; for β : 1 a.u.= 8.6393 x10-33esu)”
e) What do the authors mean by "chemical strength" ? Also some small typos should be fixed prior to resubmission, e.g.: "interactionnsare" -> "interactions are" etc.
Answer: Thanks to the reviewer for his thorough reading. The actual word was chemical stability. To remove confusion, we have replaced this word “chemical strength” with new words “Chemical stability, optical and electronic properties of molecules”. Furthermore, we have checked our manuscript again carefully to deal with the typos. All typos have been removed thoroughly in revised manuscript.
f) If possible, please define x, y and z axis in Scheme 1 (to match the labels of tensor components given in Tables 2 and 3).
Answer: It is indeed a nice suggestion and we are agreed to the comment of valuable reviewer. Therefore, x, y and z axis have been added in scheme 1 as per instructions.
We are profoundly thankful to respected reviewer for the valuable suggestions.
Reviewer 2 Report
The manuscript “Electron Donor and Acceptor Influence on the Nonlinear Optical Response of Diacetylene Functionalized Organic Material (DFOM): Density Functional Theory Calculations” by Muhammad Khalid, Muhammad Imran, Ajaz Hussain, Bakhat Ali, Farrukh Jaleel, Riaz, Hussain, Muhammad Usman Khan, Saeed Ahmed, Saba Abid, Sadia Haq, Kaynat Saleem, Shumaila Majeed, Chaudhary Jahrukh Tariq1is devoted to the DFT calculations of structure and properties of organic chromophores with dyacetylene pi-electron bridge. The authors state that the presented work is the first where the NLO properties of such molecules containing diacetylene moiety are estimated. However, previously the diacetylene oligomers were intensively studied in connection with the third-order NLO properties (see, for example, “Nonlinear optics of organic molecules and polymers, Ed. by H.S. Nalwa and S. Miyata, 2007). The authors formulate the study of the effect of the substituent in the donor- and/or acceptor end groups as one of the aims of the paper, but from my point of view this aim is not fully achieved: the authors just present the results of the calculations not clarifying the role of the position of the substituent in the D/A groups. Above all, the choice of the molecules under study is not clear: to choose a molecule with amino- and nitro- end groups as a “reference chromophore” rather than SC (which is called an “original compound”), for which one would not expect notable NLO response, seems to be more reasonable; the position of the additional moieties in D/A groups of the chromophores 1-6 at least requires additional explanation.
In my opinion the manuscript is not suitable for publication in its current form, it needs a considerable improvement.
In addition to the above remarks I would like to make several other remarks/questions:
1. In the Computational Procedure section it is pointed that CAM-B3LYP density fuctional is used for the calculation of Absorption spectra (it would be nice to have it well justified with necessary references!), but no information is given concerning the molecular polarizability calculation, though it is well known that the choice of the choice of the density functional is a key factor which may affect the results essentially.
2. The titles of the subsections in Section 3 should be more exact (for example, subsection 3.2. FMO Theory contains some concrete numerical data – E(HOMO/LUMO) and energy gap rather than the formulations of the theory)/
3. In the conclusion of subsection 3.3 “The intermolecular charge transfer, presence of extended conjugation and hyperconjugative interactions of the molecule under study can be easily evaluated on the basis of above discussion. Moreover, the capability of red shift and potential NLO character of a molecule can be associated collectively to the presence of extended conjugation and charge transfer character of compounds under study” the matter seems to be intramolecular charge transfer; the last sentence needs revision…
4. As it was mentioned already, the text of Section 3 just repeats the values of the obtained characteristics providing no insights… For example, in subsection 3.4. NLO properties the data of Tables 2 and 3 are repeated in the text; it is not clear why do the authors present (without any references!) expression (4) for alfa instead of giving the corresponding one for the first hyperpolarizability, as it is this characteristic that is the one studied in this research (according to the title of the manuscript). As beta_ijk are presented in table 3, it is worth giving a coordinate system, in particular, to analyze the data for SC (which seems to be nonplanar, though it is pointed in the Conclusion that all the molecules are planar!). The data of Table 3 are compared with that for urea “Since urea molecule is frequently employed as reference molecule [41] in literature for the comparative NLO analysis…”; this statement is out of date, it seems to be more reasonable to compare these results with those for chromophores with the structure NH2-pi-NO2, DANS (4-(dimethylamino)-4'-nitrostylbene) or azochromophores, such as DO3. The concluding remark of the subsection “From preceding discussion, it is well evident that substitution of electron D and A units in Sc changed the centrosymmetric configuration of Sc into non-centro symmetric configuration in 1-6, hence; increase the NLO response of 1-6 as compared to parent Sc.” is really evident (trivial!) and it does not have much sense.
5. It was mentioned many times that the studied molecules are planar, however SC is nonplanar on Fig.3 (and it is confirmed by table 3!) and the corresponding MEP looks strange (it has red regions which according to line 288 associates with oxygen atom which is absent in this molecule…) The last statement of the Conclusions affirms that the “research described a new paradigm forconverting centrosymmetric unit into non-centrosymmetric configurations by certain modifications in structures of D and A for designing new promising NLO compounds with huge NLO response.” However, the idea is not new; it is fully recognized by the community and confirmed by a tremendous amount of both theoretical and experimental works.
Author Response
We extremely thank the reviewer for his/her comments on the manuscript where he/she acknowledges the importance of the findings reported and conclusions reached; the useful remarks and questioning are addressed below (in yellow text). In addition we indicate revisions in the updated manuscript by a yellow highlighter in the manuscript.
Comments to the Author:
Comments and Suggestions for Authors
The manuscript “Electron Donor and Acceptor Influence on the Nonlinear Optical Response of Diacetylene Functionalized Organic Material (DFOM): Density Functional Theory Calculations” by Muhammad Khalid, Muhammad Imran, AjazHussain, Bakhat Ali, FarrukhJaleel, Riaz, Hussain, Muhammad Usman Khan, Saeed Ahmed, Saba Abid, SadiaHaq, KaynatSaleem, ShumailaMajeed, ChaudharyJahrukh Tariq1 is devoted to the DFT calculations of structure and properties of organic chromophores with dyacetylene pi-electron bridge. The authors state that the presented work is the first where the NLO properties of such molecules containing diacetylene moiety are estimated. However, previously the diacetylene oligomers were intensively studied in connection with the third-order NLO properties (see, for example, “Nonlinear optics of organic molecules and polymers, Ed. by H.S. Nalwa and S. Miyata, 2007). The authors formulate the study of the effect of the substituent in the donor- and/or acceptor end groups as one of the aims of the paper, but from my point of view this aim is not fully achieved: the authors just present the results of the calculations not clarifying the role of the position of the substituent in the D/A groups. Above all, the choice of the molecules under study is not clear: to choose a molecule with amino- and nitro- end groups as a “reference chromophore” rather than SC (which is called an “original compound”), for which one would not expect notable NLO response, seems to be more reasonable; the position of the additional moieties in D/A groups of the chromophores1-6 at least requires additional explanation. In my opinion the manuscript is not suitable for publication in its current form, it needs a considerable improvement.
Answer: We are profoundly thankful to respected reviewer for critical review. Some new discussion has been added in revised version of manuscript.
In addition to the above remarks I would like to make several other remarks/questions:
1. In the Computational Procedure section it is pointed that CAM-B3LYP density fuctional is used for the calculation of Absorption spectra (it would be nice to have it well justified with necessary references!), but no information is given concerning the molecular polarizability calculation, though it is well known that the choice of the choice of the density functional is a key factor which may affect the results essentially.
Answer: Thanks to the reviewer for his minute reading. According to the instructions of respected reviewer, the choice of CAM-B3LYP density functional used for the calculation of absorption spectra has been justified by inclusion of new references in revised draft (please see references 32-36). Furthermore, the choice of B3LYP method for the molecular polarizability calculation has been successfully proven in our previous reports. [Khan et. al., J. Cluster Sci. (2019) 1-16. ; Khan et. al., Chem. Phys. Lett. 715 (2019) 222-230.]. However, to furhter strenegthen the findings of this functional, we have again performed the calculations of polarizabilities and static first hyperpolarizabilities at the CAM-B3LYP level. The obtained results are compared with those obtained earlier using the B3LYP method. The α values are found to be with same decreasing order (6>5>3>1>4>2>Sc) in both cases. The comparison of lowest and highest α values of Sc and 6 describe that CAM-B3LYP results (233.24 and 302.38 a.u) are noticed lower as compared to the B3LYP results (248.39 and 333.40 a.u) respectively. The similar trend can be seen in case of remaining investigated compounds where B3LYP results are found higher as compared to CAM-B3LYP results. Furthermore, comparison of B3LYP results with CAM-B3LYP results implies that βtot values obtained from former (B3LYP) method is found to be greater in all investigated compounds (Sc and 1-6) as compared to those calculated from latter (CAM-B3LYP) method respectively. For instance, lowest and highest βtot values of Sc and 6 are found to be 110.56 and 24708.79 a.u from B3LYP results; while 95.57 and 11841.93 a.u from CAM-B3LYP method respectively.Thus, new Tables have been added in the revised version as suggested by the respected reviewer. The discussion has been changed accordingly in the revised manuscript. The numbering of the Tables has been modified accordingly.
2. The titles of the subsections in Section 3 should be more exact (for example, subsection 3.2. FMO Theory contains some concrete numerical data – E(HOMO/LUMO) and energy gap rather than the formulations of the theory)/
Answer: According to the guidelines provided by valuable reviewer, the title of section 3 has been changed in revised draft.
3. In the conclusion of subsection 3.3 “The intermolecular charge transfer, presence of extended conjugation and hyperconjugative interactions of the molecule under study can be easily evaluated on the basis of above discussion. Moreover, the capability of red shift and potential NLO character of a molecule can be associated collectively to the presence of extended conjugation and charge transfer character of compounds under study” the matter seems to be intramolecular charge transfer; the last sentence needs revision…
Answer: : Thanks to the reviewer for his thorough reading. We have gone through the suggested lines again carefully and sentence has been modified.
4. As it was mentioned already, the text of Section 3 just repeats the values of the obtained characteristics providing no insights… For example, in subsection 3.4. NLO properties the data of Tables 2 and 3 are repeated in the text; it is not clear why do the authors present (without any references!) expression (4) for alfa instead of giving the corresponding one for the first hyperpolarizability, as it is this characteristic that is the one studied in this research (according to the title of the manuscript). As beta_ijk are presented in table 3, it is worth giving a coordinate system, in particular, to analyze the data for SC (which seems to be nonplanar, though it is pointed in the Conclusion that all the molecules are planar!).The data of Table 3 are compared with that for urea “Since urea molecule is frequently employed as reference molecule [41] in literature for the comparative NLO analysis…”; this statement is out of date, it seems to be more reasonable to compare these results with those for chromophores with the structure NH2-pi-NO2, DANS (4-(dimethylamino)-4'-nitrostylbene) or azochromophores, such as DO3. The concluding remark of the subsection “From preceding discussion, it is well evident that substitution of electron D and A units in Sc changed the centrosymmetric configuration of Sc into non-centro symmetric configuration in 1-6, hence; increase the NLO response of 1-6 as compared to parent Sc.” is really evident (trivial!) and it does not have much sense.
Answer: We are much obliged to the comments from respected reviewer. According to the instructions of valuable reviewer, the subsection NLO properties discussion has been modified. The references of expression (4) for alfa have been added in the revised draft (please see references 50-52). A coordinate system indicating the x, y and z-axis has been added and highlighted in scheme 1. Furthermore, for comparative NLO analysis, the reviewer suggested chromophores 4-(dimethylamino)-4'-nitrostylbene (DANS) has been utilized in revised version. We have performed the DFT computations to calculate the βtot value of DANS at B3LYP (gas phase), CAM-B3LYP (gas phase) and CAM-B3LYP (solvent; DMSO) level of theory and compared its results with our investigated molecules results in revised version. The comparison of designed molecules 1-6 results with DANS results indicates that designed molecules 1-6 exhibit 1.06-2.21 times greater NLO response than DANS calculated through B3LYP (gas phase), 1.41- 2.20 times greater NLO response than DANS measured from CAM-B3LYP (gas phase) and 1.28- 1.99 times greater NLO response than DANS calculated through CAM-B3LYP (solvent DMSO) methods respectively. The NLO comparative analysis has been revised and modified in revised manuscript. The concluding remark of the subsection has been altered as per instructions of respected reviewer.
5. It was mentioned many times that the studied molecules are planar, however SC is nonplanar on Fig.3 (and it is confirmed by table 3!) and the corresponding MEP looks strange (it has red regions which according to line 288 associates with oxygen atom which is absent in this molecule…) The last statement of the Conclusions affirms that the “research described a new paradigm for converting centrosymmetric unit into non-centrosymmetric configurations by certain modifications in structures of D and A for designing new promising NLO compounds with huge NLO response.” However, the idea is not new; it is fully recognized by the community and confirmed by a tremendous amount of both theoretical and experimental works.
Answer: We are deeply grateful to respected reviewer. In MEP diagram, the red color of SC molecule (without oxygen atom) is different from designed molecules red color (containing oxygen atom). The pure red color indicating the site for possible electrophilic attack can be seen on designed molecules 1-6. However in SC, the light red color appeared on both opposite ends of molecule due to lone pair of electron available on nitrogen atom. We have changed the few words in discussion of MEP section. Furthermore we are agreed with the comment of valuable reviewer, so, the last statement of the conclusion section has been changed and modified according to the guidelines of respected reviewer.
We are profoundly thankful to respected reviewer for the valuable suggestions.
Round 2
Reviewer 1 Report
The suggested changes have been implemented properly.
Author Response
Reviewer: 1
We are deeply grateful to the reviewer for taking the extensive time to provide quite valuable comments, suggestions and encouraging evaluation.
Comments to the Author:
The suggested changes have been implemented properly.
Answer: We are profoundly thankful to respected reviewer for positive recommendation.
Reviewer 2 Report
The authors have revised the manuscript seriously mainly providing the results of some additional calculations (those with CAM-B3LYP density functional and with the account of solvent effect). However, the discussion of the results still can be improved: it is very verbose, the text contains many repeatitions (in particular, the values given in the tables are again given in the text) providing no insights...
The remark concerning the choice of the molecules under study is left unanswered: the comment given in the first report “to choose a molecule with amino- and nitro- end groups as a “reference chromophore” rather than SC (which is called an “original compound”), for which one would not expect notable NLO response, seems to be more reasonable; the position of the additional moieties in D/A groups of the chromophores 1-6 at least requires additional explanation” may be repeated.When comparing the results of the first hyperpolarizability calculations, the authors should have more clear position concerning the choice of the density functional: on the one hand, they acknowledge that the use of B3LYP results in the jverestemsted beta values, on the other hand they write (321-322): “From preceding discussion, it is well evident that gas phase results from B3LYP method are found better as compared to CAM-B3LYP gas phase results.” This statement looks at least strange…
In the revised version the authors have added the comparison with DANS but still remained the comparison with urea which seems to have little sense (of course, it gave the authors a possibility to state that “The βtot value of 1-6 were observed as 275-574 times, 312 177-275 times, and 374-582 times greater than the βtot value of urea molecule calculated through 313 B3LYP (gas), CAM-B3LYP (gas) and CAM-B3LYP (DMSO) methods respectively”!)
Author Response
Reviewer 2
We extremely thank the reviewer for his/her comments on the manuscript where he/she acknowledges the importance of the findings reported and conclusions reached; the useful remarks and questioning are addressed below (in yellow text). In addition, we indicate revisions in the updated manuscript by a yellow highlighter in the manuscript.
Comments to the Author:The authors have revised the manuscript seriously mainly providing the results of some additional calculations (those with CAM-B3LYP density functional and with the account of solvent effect).
Answer: We are profoundly thankful to respected reviewer. We are motivated as the referee has considered this manuscript seriously revised with results of some additional calculations.
However, the discussion of the results still can be improved: it is very verbose, the text contains many repetitions (in particular, the values given in the tables are again given in the text) providing no insights.
Answer: Thanks to the reviewer for detailed study of the manuscript. According to the instructions of valuable reviewer, we have checked our manuscript again carefully to deal with the verbose discussion of results. The redundant values have been removed thoroughly in revised manuscript.
The remark concerning the choice of the molecules under study is left unanswered: the comment given in the first report “to choose a molecule with amino- and nitro end groups as a reference chromophores rather than SC (which is called an “original compound”), for which one would not expect notable NLO response, seems to be more reasonable; the position of the additional moieties in D/A groups of the chromophores 1-6 at least requires additional explanation” may be repeated.
Answer: In fact, this research work was started by the inspiration from a previous report of Pachfule et al. wherein they synthesis of diacetylene-bridged covalent organic frameworks (COFs) TP-BDDA and reported its interesting features (see reference # 23). In BDDA architecture, two phenyl rings are connected with each other through diacetylene-bridge and two amino groups are present at both ends of phenyl rings leading to a centrosymmetric configuration, which was considered as a parent compound (Sc).
In our study, a series of derivatives-based donor−π−acceptor have been designed by structural tailoring for developing non-centrosymmetric configurations. For the sake of clarity, the notation “SC (original compound)” has been changed to reference chromophore (R) in the revised manuscript as suggested by the valuable reviewer. Moreover, the notation "Sc" has also been replaced by R in supporting information. Further explanation regarding position of the additional moieties in D/A groups of the chromophores 1-6 has been added in the revised draft.
When comparing the results of the first hyperpolarizability calculations, the authors should have more clear position concerning the choice of the density functional: on the one hand, they acknowledge that the use of B3LYP results in the overestimated beta values, on the other hand they write (321-322): “From preceding discussion, it is well evident that gas phase results from B3LYP method are found better as compared to CAM-B3LYP gas phase results.” This statement looks at least strange…
Answer: Thanks to the reviewer for his thorough reading. Actually we had performed our calculations with B3LYP level of theory. But during first revision, first reviewer guided us to include these lines “The conventional global hybrid DFT functionals, like B3LYP, tend to overestimate first and second hyperpolarizabilities, and in some cases can provide incorrect trends even within structurally related series [16, 17]. Therefore, to get an idea about the correct trends that are recovered using long-range corrected functionals, calculations of polarizabilities and first hyperpolarizabilities have been performed at the CAM-B3LYP level of theory and results are compared with those obtained using the B3LYP method” along with the cited references. So these lines and references were added according to the instructions of first reviewer. First reviewer was also interested to see the results of polarizabilities and first hyperpolarizabilities at CAM-B3LYP level of theory. They had the view to perform first and second hyperpolarizabilities with CAM-B3LYP functional and compared your B3LYP results with CAM-B3LYP result. Following the instructions of first respected reviewer, we performed the calculations and found gas phase B3LYP results better than CAM-B3LYP results as mentioned in the draft.
In the revised version the authors have added the comparison with DANS but still remained the comparison with urea which seems to have little sense (of course, it gave the authors a possibility to state that “The βtot value of 1-6 were observed as 275-574 times, 312 177-275 times, and 374-582 times greater than the βtot value of urea molecule calculated through 313 B3LYP (gas), CAM-B3LYP (gas) and CAM-B3LYP (DMSO) methods respectively”!)
Answer: Thanks to the reviewer for detailed study of the manuscript. The discussion on comparison with urea molecule has been removed in the revised draft.
We are profoundly thankful to respected reviewer for the valuable suggestions.